# Spatial Understanding from Videos: Structured Prompts Meet Simulation Data

**Haoyu Zhang**[1,2], **Meng Liu**[3,4*], **Zaijing Li**[1,2], **Haokun Wen**[1],
**Weili Guan**[1], **Yaowei Wang**[1,2], **Liqiang Nie**[1*]

[1]Harbin Institute of Technology (Shenzhen)   [2]Pengcheng Laboratory
[3]Shandong Jianzhu University   [4]Zhongguancun Academy

## Abstract

Visual-spatial understanding, the ability to infer object relationships and layouts from visual input, is fundamental to downstream tasks such as robotic navigation and embodied interaction. However, existing methods face spatial uncertainty and data scarcity, limiting the 3D spatial reasoning capability of pre-trained vision-language models (VLMs). To address these challenges, we present a unified framework for enhancing 3D spatial reasoning in pre-trained VLMs without modifying their architecture. This framework combines SpatialMind, a structured prompting strategy that decomposes complex scenes and questions into interpretable reasoning steps, with ScanForgeQA, a scalable question-answering dataset built from diverse 3D simulation scenes through an automated construction process designed for fine-tuning. Extensive experiments across multiple benchmarks demonstrate the individual and combined effectiveness of our prompting and fine-tuning strategies, and yield insights that may inspire future research on visual-spatial understanding.

## 1 Introduction

Visual-spatial understanding, the ability to infer spatial relationships and the layout of objects from visual input, is a core component of human perception [1, 2, 3]. From a single image, human observers can intuitively estimate distances, relative sizes, and even infer occluded structures. As intelligent systems become increasingly embedded in real-world applications such as autonomous driving [4, 5, 6], robotic navigation [7, 8, 9], and augmented reality [10, 11, 12], it becomes crucial to endow models with similar spatial reasoning capabilities for robust perception and interaction.

Unfortunately, a single image is inherently limited in capturing the complexity of real-world 3D scenes, constraining its utility in practical scenarios [13, 14, 15]. To address this, point clouds have become a mainstream representation for 3D scene understanding due to their ability to encode rich geometric information [16, 17]. Yet, generating high-quality point clouds typically requires expensive sensors and incurs significant computational overhead, limiting scalability and accessibility.

These limitations motivate the pursuit of vision-only solutions that operate on scanning videos or multi-view images of scenes. Such approaches offer a more human-like and scalable pathway to spatial understanding [18]. However, performing 3D spatial reasoning from scanning videos presents two significant challenges: **(1) Spatial Uncertainty.** In the absence of explicit depth information, models must infer 3D structure from inherently limited 2D observations. This process is further complicated by occlusions, perspective distortions, and texture ambiguities, all of which introduce significant spatial uncertainty. Effectively addressing this challenge demands multi-step logical reasoning across frames to reconstruct coherent spatial layouts. **(2) Data Scarcity.** Existing datasets for this task are limited in both scale and diversity, restricting the ability of vision-language models

---

*Corresponding authors.

39th Conference on Neural Information Processing Systems (NeurIPS 2025).

(VLMs) to acquire robust spatial knowledge and perceptual capabilities. Moreover, these datasets involve scans of real-world scenes, which leads to poor scalability. This highlights the need for scalable and extensible data sources to support effective spatial reasoning in VLMs.

To address these challenges, we propose a dual approach for enhancing 3D spatial reasoning in pre-trained VLMs, without modifying their underlying architecture. First, we introduce **SpatialMind**, a structured Chain-of-Thought (CoT) prompting strategy that guides VLMs through step-by-step reasoning over spatial relationships. Second, we present **ScanForgeQA**, a large-scale synthetic question-answering (QA) dataset constructed from diverse 3D simulation scenes using an automated generation pipeline. Fine-tuning VLMs on this dataset equips them with spatial commonsense knowledge, significantly improving their generalization to unseen spatial layouts. We have validated our approach through extensive experiments across multiple benchmarks. Results demonstrate the individual and combined effectiveness of our prompting and fine-tuning strategies, and yield insights that may inspire future research on visual-spatial understanding.

Our contributions are summarized as follows:

- We introduce SpatialMind, a spatial prompting strategy that decomposes spatial reasoning into structured steps, enabling pre-trained VLMs to perform multi-step inference over spatial relationships from visual input alone.

- We develop a scalable dataset generation pipeline to construct ScanForgeQA, a synthetic spatial question-answering dataset that enables VLMs to acquire spatial commonsense through fine-tuning.

- Experimental results validate the effectiveness and generalizability of both SpatialMind and ScanForgeQA, with their combination achieving further gains and providing valuable insights for future research.

## 2   Related Work

**2D Image Spatial Understanding** focuses on modeling spatial relationships among objects within the 2D image. Most existing models are trained on 2D images paired with textual descriptions, which offer limited cues about 3D structure. Consequently, their capacity for spatial reasoning remains constrained. To mitigate this, several approaches, such as SpatialVLM [13], SpatialRGPT [14], and SpatialBot [15], have been proposed. These methods enhance the spatial understanding by fine-tuning models on datasets specifically designed for spatially grounded QA tasks. To enable more comprehensive evaluation, recent studies [19, 20, 21, 22, 23] have introduced hierarchical benchmarks that assess models across varying levels of spatial reasoning complexity. Parallel efforts have explored more explicit forms of spatial interaction [24, 25]. For example, point-based methods [26, 27] interpret spatial instructions by predicting specific target points. Building on this trend, SpatialCoT [28] proposes a two-stage strategy that aligns multimodal inputs with spatial coordinates and incorporates CoT reasoning to better address complex embodied tasks. Despite these advancements, model performance often degrades in complex real-world 3D environments, highlighting the limitations of 2D-based approaches in representing complex 3D scenes.

**3D Indoor Spatial Understanding** focuses on enabling intelligent agents to identify object positions and infer their spatial relationships within enclosed environments, thereby supporting both object manipulation and interactive scene comprehension. Early 3D models are trained on standard indoor datasets [29, 30, 31, 32, 33, 34, 35] using point clouds to facilitate downstream tasks like 3D object detection and instance segmentation [36, 37, 38, 39] and primarily focus on object-level geometry and appearance features [40, 41, 42, 43]. More recent work extends this focus to complex indoor scenes, emphasizing inter-object spatial relationships and holistic scene-level understanding. To address challenges such as geometric complexity and annotation sparsity, many of these models employ cross-modal strategies that combine point cloud data with auxiliary multi-view 2D images [16, 17, 44]. Inspired by the way humans perceive spatial layouts through vision alone, emerging research [1, 45, 18] has begun to explore purely vision-based approaches to 3D spatial understanding. These methods rely solely on visual inputs, such as scanning videos, without requiring explicit 3D priors like point clouds. This line of work offers a more practical and scalable alternative for real-world deployment. In this context, we further investigate whether purely vision-based inputs can provide a more effective solution for indoor scene understanding.

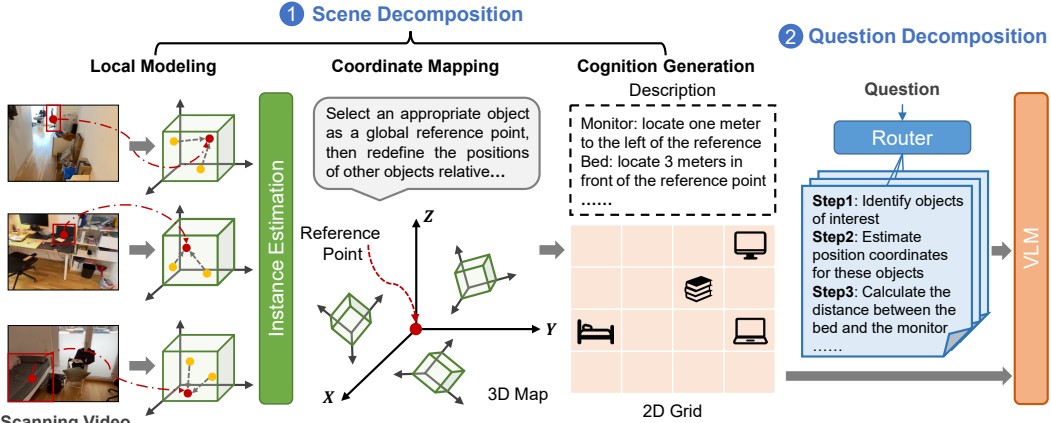

Figure 1: Illustration of our SpatailMind prompting strategy.

# 3 SpatialMind Prompting Strategy

As shown in Figure 1, our SpatialMind prompting strategy consists of two main components: **1) Scene Decomposition**, where the 3D scene depicted in the video is transformed into multiple different representations; and **2) Question Decomposition**, in which the question is broken down into a sequence of fine-grained reasoning steps. Further details can be found in **Appendix D**.

## 3.1 Scene Decomposition

The scene decomposition process includes three sequential steps: local modeling, coordinate mapping, and cognition generation.

**Local Modeling**. The first step processes scanning video frames to extract object instances and their relative spatial configurations within localized coordinate systems. To handle scene complexity and reduce the search space, we leverage GPT-4o[2] to identify all objects mentioned across the questions associated with a given scene, using them as candidate targets. For each frame $i$, we prompt VLMs to detect a subset of objects $\{c_{ij}\}$ from the candidate targets and estimate their positions $\mathbf{p}_{ij}^{\text{local}} \in \mathbb{R}^3$. These positions are defined relative to a randomly selected reference object (i.e., origin) within the same frame, forming a local 3D map:

$$\mathcal{L}_i = \left\{ (c_{ij}, \mathbf{p}_{ij}^{\text{local}}) \mid j = 1, \ldots, n_i \right\}, \tag{1}$$

where $n_i$ denotes the number of objects in frame $i$. Because each video frame captures only a limited field of view, the same object may appear across multiple frames from different perspectives. Thus, this step focuses on accurate per-frame object detection and spatial localization, laying the foundation for subsequent alignment in a global coordinate system.

**Coordinate Mapping**. To integrate spatial information across video frames, this step transforms all locally detected object positions into a unified global coordinate system. The global origin is defined by selecting the reference object in the first frame. To estimate motion between frames, we prompt the VLM to infer the relative rotation and translation between adjacent frames. These relative transformations are accumulated sequentially to compute each frame's transformation $\mathbf{T}_i$ with respect to the global coordinate system:

$$\mathbf{T}_i = \prod_{k=1}^{i} \begin{bmatrix} \mathbf{R}_{k,k-1} & \mathbf{t}_{k,k-1} \\ \mathbf{0} & 1 \end{bmatrix}, \tag{2}$$

where $\mathbf{R}_{k,k-1}$ and $\mathbf{t}_{k,k-1}$ denote the relative rotation and translation from frame $k-1$ to frame $k$, respectively. This accumulated approach provides more stable and accurate alignment than directly estimating each frame's absolute pose. Using these transformations, each object's local coordinates are converted into global coordinates via homogeneous transformation:

$$\begin{bmatrix} \mathbf{p}_{ij}^{\text{global}} \\ 1 \end{bmatrix} = \mathbf{T}_i \cdot \begin{bmatrix} \mathbf{p}_{ij}^{\text{local}} \\ 1 \end{bmatrix}, \tag{3}$$

---

[2]`https://openai.com/index/hello-gpt-4o/`.

where $\mathbf{p}_{ij}^{\text{global}}$ denotes the global coordinates of the object $j$ in the frame $i$. This step ensures that all detected objects across frames are positioned consistently within the same 3D space. Since objects may appear in multiple frames under different perspectives, we merge duplicate detections based on spatial proximity and semantic consistency via prompting. The result is a global 3D map of the scene:

$$\mathcal{G} = \left\{ (c_k, \mathbf{p}_k^{\text{global}}) \right\}_{k=1}^{N}, \tag{4}$$

where $N$ is the total number of all object instances in the entire scene. This map serves as a unified spatial abstraction that captures the overall layout from egocentric scanning videos.

**Cognition Generation**. Beyond constructing a 3D map, we explore two additional formats for representing scene structure: a 2D spatial grid and natural language descriptions. We define a regular 2D grid over the global scene, typically aligned with the $XY$-plane. Each grid cell corresponds to a fixed real-world area (e.g., 1 meter per cell, denoted by cell size $s$). Each object $c_k$ is mapped to a discrete grid location $(i_k, j_k)$:

$$(i_k, j_k) = \left( \left\lfloor \frac{x_k}{s} \right\rfloor, \left\lfloor \frac{y_k}{s} \right\rfloor \right), \tag{5}$$

where $(x_k, y_k)$ are the horizontal components of the object's global position $\mathbf{p}_k^{\text{global}}$. In parallel, we generate natural language descriptions of object locations relative to a designated reference point. Using prompting, the model produces statements such as {"monitor": "locate 1 meter to the left of the reference point"}. These descriptions serve as a human-interpretable form of spatial cognition, bridging visual perception and symbolic reasoning.

## 3.2 Question Decomposition

Different types of spatial questions require distinct reasoning strategies [46]. To accommodate this diversity, we first categorize questions into several types (e.g., object size, relative distance, and relative direction). For each category, we design a dedicated reasoning procedure using GPT-4o, followed by human verification to ensure correctness and interpretability. For instance, consider a question from the "relative distance" category: *Among the refrigerator, window, and microwave, which object is closest to the door?* The reasoning process for this type follows four structured steps: 1) Identify all mentioned objects, 2) Estimate the spatial coordinates of all relevant objects, 3) Compute the pairwise distances between the door and each candidate object, and 4) Select the object with the minimum distance as the answer. During inference, the system correspondingly selects the appropriate reasoning procedure based on the identified question type.

To perform 3D spatial reasoning, we feed the VLMs with the input scanning video, one form of scene representation (e.g., 3D map, 2D grid, or textual position descriptions), the question, and the corresponding step-by-step reasoning plan. To assess which scene representation format is most interpretable for VLMs, we have conducted comparative experiments, as shown in Figure 3.

## 4  ScanForgeQA Dataset Construction

The construction of the **ScanForgeQA** dataset involves a three-stage pipeline, illustrated in Figure 2. These stages are: **1) Scene Construction**, where single-room 3D environments are created; **2) Scan Creation**, in which egocentric videos are simulated by scanning through the constructed scenes; and **3) QA Generation**, where textual question-answering pairs are automatically generated based on object annotations and the spatial layout of each scene.

### 4.1  Scene Construction

To ensure diversity and richness in single-room scene collection, we adopt two parallel strategies:

**Separation**. We modify existing scene datasets to leverage available resources effectively. Specifically, we utilize the 3D-FRONT dataset [47], which contains 6,813 multi-room scenes furnished with diverse 3D objects and annotated with detailed layout semantics and high-quality textures. Since our focus is on single-room environments, we disassemble each multi-room scene into individual rooms. For each scene, we isolate and load one room at a time, along with its corresponding ceiling and walls, and save it as an independent instance. This disassembly process yields 44,427 single-room scenes. We further filter out uncommon room types (e.g., garage, auditorium) and those lacking sufficient

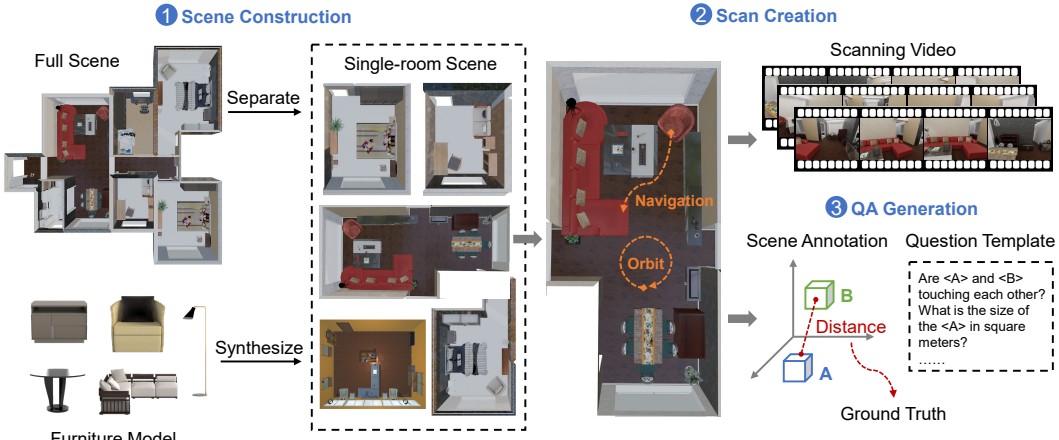

Figure 2: The pipeline of ScanForgeQA data construction.

object content (e.g., aisle, stairwell). The final dataset consists of 34,116 single-room scenes across six common categories: bedroom, kitchen, bathroom, living room, dining room, and storage room.

**Synthesis**. To introduce additional diversity and originality, we synthesize novel room layouts using a LLM-guided generation approach. Specifically, we adopt HoloDeck [48], a 3D generation framework that leverages LLMs to parse natural language prompts, retrieve matching assets from large-scale 3D object repositories such as Objaverse [49], and optimize their spatial arrangement to form semantically meaningful scenes. To drive the generation process, we first use GPT-4o to create diverse textual descriptions for various room types. For example, a bedroom may be described as: *"A bedroom with a bed, window, armchair, and wardrobe"*. We define eight room categories, including two additional types—office and store—and generate 20 distinct descriptions for each. These prompts are fed into HoloDeck to produce corresponding room layouts, with human verification to ensure spatial plausibility and realism. This synthesis process yields 160 additional single-room scenes.

## 4.2 Scan Creation

To simulate egocentric scanning videos from the constructed single-room scenes, we implement a scanning procedure using the Unity engine[3]. Each scene is scanned using two complementary strategies designed to emulate natural human visual exploration:

**Orbit Scan**. We define a circular trajectory centered in the room at a height of approximately 1.5 meters, corresponding to typical adult eye level. The circle's diameter is set to two-thirds of the shorter side of the room. The camera is randomly initialized at a point on this path and moves along the circle either clockwise or counterclockwise. An image is captured every 5 degrees of rotation, resulting in 72 frames per orbit scan. This strategy provides a comprehensive 360-degree panoramic view of the scene.

**Navigation Scan**. To simulate movement through the environment, we label navigable ground regions based on object categories and generate a navigation mesh using the *NavMesh Baking* API. We randomly select two objects as the navigation start and end points and compute the shortest path between them on the mesh. Among the candidate paths, the two longest are chosen for scanning to achieve a more complete coverage of the scene. For each path, the camera first performs a 360-degree rotation at the starting point, capturing an image every 12 degrees (30 images total). It then traverses the path toward the destination, during which 12 frames are uniformly sampled. Upon arrival, another 360-degree rotation is performed, again capturing 30 images. In total, 72 frames are recorded per path. Due to the limited size of indoor environments, rotational movement yields more visual variation than translation; hence, fewer frames are captured during motion.

---

[3]https://unity.com/.

Table 1: Comparison of 3D QA Datasets

| Dataset | Source | Scenes | Format | QAs | Scalability |
|---|---|---|---|---|---|
| SPARTUN3D | 3RScan | 478 | Point Cloud | 133K | Hard |
| MSQA | ScanNet, 3RScan, ARKitScenes | 1.7K | Point Cloud | 251K | Hard |
| 3D-LLM | ScanNet, HM3D | 1.2K | Point Cloud | 300K | Hard |
| ScanForgeQA | Simulation/Synthesis | 34K | Scan Video | 925K | Easy |

## 4.3 QA Generation

To generate diverse supervised fine-tuning (SFT) data and enhance the 3D spatial reasoning capabilities of existing VLMs, we define three categories of question types: attribute estimation, spatial reasoning, and hypothesis analysis. These categories encompass both quantitative and qualitative dimensions, and cover both open-set and closed-set scenarios. Below, we describe each category in detail, along with the methodology for deriving corresponding ground-truth answers.

**Attribute Estimation**. This type focuses on static properties of objects and scenes, such as *object count* ("How many chairs are in the room?"), *object size* ("What is the length of the longest side of the refrigerator in meters?"), *room size* ("What is the size of this room in square meters?"), and *room type* ("Based on the object layout, what is the most likely type of room (e.g., kitchen)?"). Ground-truth answers for these questions are directly derived from 3D scene annotations and object metadata provided in the dataset.

**Spatial Reasoning**. This category targets inter-object spatial relationships, requiring models to infer positional and geometric properties such as distance, orientation, and contact. Representative question types include: *relative distance* ("Which of these objects (refrigerator, couch, ceiling light) is closest to the TV?"), *absolute distance* ("What is the distance between the couch and the table in meters?"), *relative direction* ("If I am standing by sofa and facing the table, which side is the trash can on?"), and *contact relationship* ("Is there a gap between the bed and the headboard?"). For distance-related questions, we compute Euclidean distances between object centroids in the global 3D coordinate space. For contact relationships, object dimensions are also considered to determine physical adjacency. To resolve relative direction,

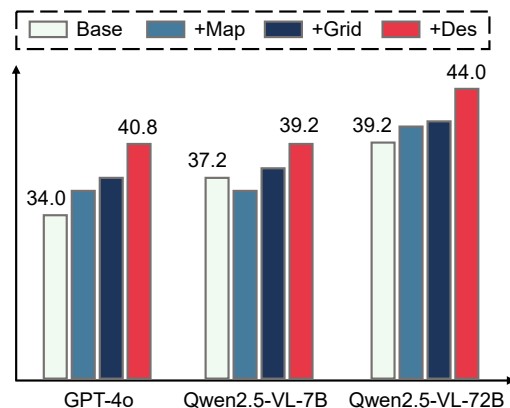

Figure 3: Effects of different scene expression.

we define an object's front as the side oriented toward the room center. Angular sectors are divided clockwise into four directional categories: right (45°–135°), back (135°–225°), left (225°–315°), and front (315°–45°). For example, an object located at 80° relative to the reference point is classified as being on the right.

**Hypothesis Analysis**. This category introduces conditional reasoning under hypothetical scenarios, often requiring geometric and commonsense inference. A typical example is *operation feasibility* ("Considering only object dimensions, is it feasible to place the television on the table?"). Feasibility is determined by comparing object dimensions. For stacking, the movable object's length and width must be smaller than those of the supporting surface. For embedding (e.g., fitting an item into a drawer), the object's height must also fall within the bounds of the specific container's volume.

A comparison with existing 3D QA datasets (e.g., SPARTUN3D [50], MSQA [51], and 3D-LLM [17]) is presented in Table 1. The full ScanForgeQA dataset includes 34,276 single-room scenes, 103K simulated video scans, and 925K question-answering pairs for training. Leveraging synthetic environments allows scalable and controlled data generation across diverse spatial scenarios. Additional implementation details are provided in the **Appendix C**.

Table 2: Performance comparison on VSI-Bench. † indicates results on VSI-Bench (tiny) set.

| Method | Obj. Count | Abs. Dist. | Obj. Size | Room Size | Rel. Dist. | Rel. Dir. | Route Plan | Appr. Order | Avg | Δ |
|---|---|---|---|---|---|---|---|---|---|---|
| **Close-source** | | | | | | | | | | |
| Human Level† | 94.3 | 47.0 | 60.4 | 45.9 | 94.7 | 95.8 | 95.8 | 100.0 | 79.2 | - |
| Gemini-1.5 Pro† | 49.6 | 28.8 | 58.6 | 49.4 | 46.0 | 48.1 | 42.0 | 68.0 | 48.8 | - |
| Gemini-1.5 Pro | 56.2 | 30.9 | 64.1 | 43.6 | 51.3 | 46.3 | 36.0 | 34.6 | 45.4 | - |
| +SpatialMind | 63.9 | 51.8 | 70.2 | 47.3 | 56.3 | 45.9 | 42.6 | 44.3 | 52.8 | ↑ **7.4%** |
| GPT-4o | 46.2 | 5.3 | 43.8 | 38.2 | 37.0 | 41.3 | 31.5 | 28.5 | 34.0 | - |
| +SpatialMind | 40.0 | 27.1 | 62.7 | 40.9 | 41.0 | 39.6 | 37.1 | 38.5 | 40.8 | ↑ **6.8%** |
| **Open-source** | | | | | | | | | | |
| Struct2D | 46.0 | 34.7 | 56.4 | 42.6 | 35.1 | 44.9 | 33.5 | - | 41.9 | - |
| InternVL2-8B | 23.1 | 28.7 | 48.2 | 39.8 | 36.7 | 30.7 | 29.9 | 39.6 | 34.6 | - |
| +SpatialMind | 35.8 | 28.9 | 49.7 | 44.4 | 37.2 | 34.8 | 35.1 | 45.5 | 38.9 | ↑ **4.3%** |
| +ScanForgeQA | 45.3 | 33.4 | 54.8 | 45.0 | 41.1 | 36.1 | 33.4 | 43.0 | 41.5 | ↑ **6.9%** |
| +Both | 47.0 | 32.8 | 53.2 | 46.6 | 39.8 | 36.8 | 37.9 | 47.5 | 42.7 | ↑ **8.1%** |
| InternVL2-40B | 34.9 | 26.9 | 46.5 | 31.8 | 42.1 | 32.2 | 34.0 | 39.6 | 36.0 | - |
| +SpatialMind | 36.4 | 30.0 | 49.1 | 41.8 | 43.8 | 36.1 | 35.6 | 50.0 | 40.4 | ↑ **4.4%** |
| +ScanForgeQA | 51.0 | 29.2 | 52.7 | 38.1 | 47.2 | 36.4 | 35.9 | 47.6 | 42.3 | ↑ **6.3%** |
| +Both | 52.2 | 30.5 | 54.4 | 41.0 | 50.5 | 37.0 | 40.2 | 50.3 | 44.5 | ↑ **8.5%** |
| Qwen2.5-VL-7B | 40.3 | 22.2 | 50.1 | 38.9 | 38.0 | 40.7 | 31.4 | 35.9 | 37.2 | - |
| +SpatialMind | 45.1 | 25.2 | 52.1 | 41.4 | 38.7 | 41.6 | 34.7 | 34.5 | 39.2 | ↑ **2.0%** |
| +ScanForgeQA | 53.2 | 30.5 | 56.8 | 44.9 | 42.3 | 44.0 | 37.3 | 37.7 | 43.3 | ↑ **6.1%** |
| +Both | 55.0 | 29.5 | 57.3 | 44.0 | 43.5 | 44.3 | 38.3 | 39.2 | 43.9 | ↑ **6.7%** |
| Qwen2.5-VL-72B | 37.9 | 28.6 | 57.4 | 49.8 | 45.5 | 38.4 | 20.6 | 35.4 | 39.2 | - |
| +SpatialMind | 42.3 | 32.0 | 61.7 | 53.8 | 48.2 | 43.9 | 30.4 | 39.3 | 44.0 | ↑ **4.8%** |
| +ScanForgeQA | 45.2 | 32.7 | 63.3 | 52.4 | 50.1 | 41.7 | 32.8 | 40.2 | 44.8 | ↑ **5.6%** |
| +Both | 48.6 | 34.4 | 68.9 | 54.7 | 53.4 | 43.9 | 30.1 | 42.7 | 47.1 | ↑ **7.9%** |

Table 3: Performance comparison on the EM-EQA subset of OpenEQA and the validation set of ScanQA and SQA3D.

| Method | OpenEQA Acc/Score | ScanQA BLEU-1 | SQA3D EM-1 |
|---|---|---|---|
| Qwen2.5-VL-7B | 50.1/3.1 | 32.5 | 17.2 |
| +SpatialMind | 53.7/3.2 | 33.1 | 19.8 |
| +ScanForgeQA | 56.2/3.3 | 34.8 | 23.3 |
| +Both | 58.6/3.5 | 37.9 | 24.5 |
| Qwen2.5-VL-72B | 53.8/3.2 | 35.4 | 34.8 |
| +SpatialMind | 55.7/3.2 | 38.0 | 39.2 |
| +ScanForgeQA | 59.1/3.4 | 42.5 | 43.0 |
| +Both | 60.4/3.4 | 44.1 | 46.3 |

Table 4: Effects of different fine-tuning data and prompting strategy.

| Method | Room Size | Avg |
|---|---|---|
| Qwen2.5-VL-7B | 38.9 | 37.2 |
| +SQA3D | 38.8 | 38.9 |
| +ScanQA | 38.5 | 39.1 |
| +ScanForgeQA | 44.9 | 43.3 |
| Qwen2.5-VL-72B | 49.8 | 39.2 |
| +CoT-Question | 50.6 | 41.3 |
| +CoT-Scene | 52.1 | 42.7 |
| +SpatialMind | 53.8 | 44.0 |

## 5 Experiments

The experimental settings (including benchmarks, baselines, etc.) and more experimental results can be found in the **Appendix A** and **B**.

### 5.1 Performace Comparison

We investigated the following five key questions to assess our approach:

**Q1: Which scene representation format is most interpretable by VLMs?** Figure 3 presents a performance comparison across different representation formats: no additional spatial context (Base), inclusion of a 3D map (+Map), a 2D grid (+Grid), and object-centric textual descriptions (+Des). Across all models, a consistent trend emerges: the +Des variant outperforms others, followed by

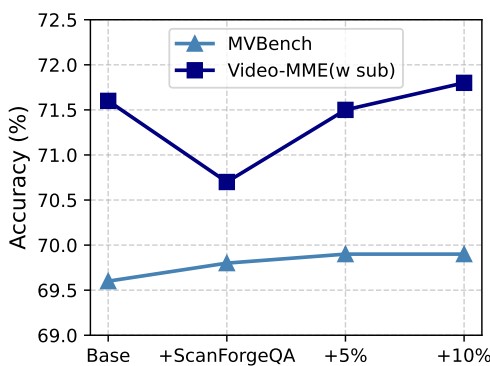
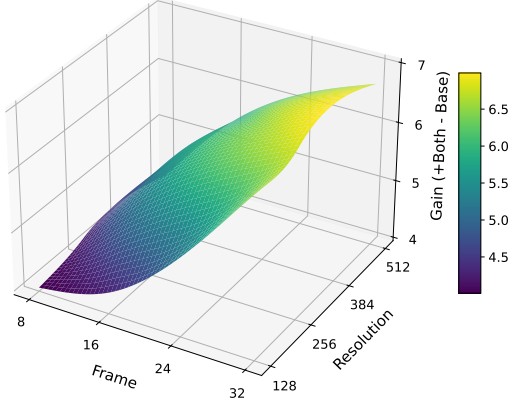

Figure 4: Performance of Qwen2.5-VL-7B on MVBench and Video-MME.

Figure 5: Ablation study of Qwen2.5-VL-7B under varying numbers of frames and resolution.

+Grid, while +Map yields the least improvement. These results suggest that current VLMs are more adept at interpreting one-dimensional textual descriptions than high-dimensional structured spatial formats. Consequently, we adopted the textual description format in subsequent experiments as the default scene representation.

**How do SpatialMind and ScanForgeQA impact VLM performance?** As shown in Table 2, we progressively applied the SpatialMind prompting strategy and the ScanForgeQA fine-tuning data across a range of VLMs, varying in architectures, parameter size, and openness (including both open- and closed-source models). The results reveal three key findings: 1) Both SpatialMind prompting and ScanForgeQA fine-tuning consistently improve visual-spatial understanding across models. This includes large-scale proprietary models such as Gemini-1.5 Pro [4] and GPT-4o, demonstrating the effectiveness and generalizability of our approaches. 2) Model size affects the relative benefit of prompting versus fine-tuning. Larger models (e.g., 72B) benefit more from prompting, which enhances their reasoning capabilities, while smaller models (e.g., 7B) show greater improvements through fine-tuning. For instance, Qwen2.5-VL-7B gains 6.1% from fine-tuning, compared to only 2.0% from prompting. 3) Humans and VLMs exhibit complementary strengths. Human participants excel in qualitative tasks (e.g., achieving 100% accuracy on the *Appearance Order* task) but perform poorly on precise quantitative estimations (e.g., *Object Size*). In contrast, VLMs show strong quantitative reasoning ability and, in some cases, even surpass human-level performance. This contrast underscores the potential of VLMs to complement human perception in spatial tasks.

**Can combining prompting and fine-tuning yield further gains?** To assess whether SpatialMind and ScanForgeQA provide complementary benefits, we applied the SpatialMind prompting strategy to models that have already been fine-tuned on the ScanForgeQA dataset. The results, reported in the "+Both" row of Table 2, show consistent performance improvements across all evaluated models. These findings confirm that the two approaches are complementary.

**Does the improvement generalize to other spatial benchmarks?** To assess the generalizability of our framework, we conducted evaluations on multiple benchmarks, including OpenEQA [52], ScanQA [53], and SQA3D [54]. As shown in Table 3, both SpatialMind prompting and ScanForgeQA fine-tuning lead to consistent performance gains across all benchmarks. These results validate the robustness of our approach and confirm its applicability across diverse spatial tasks and datasets.

**Does fine-tuning affect performance on other tasks?** To investigate whether enhancing visual-spatial capabilities via fine-tuning adversely impacts a model's general performance, we conducted evaluations on MVBench [55] and Video-MME [56], two broad multi-task video benchmarks. As shown in Figure 4, fine-tuning with ScanForgeQA slightly improves performance on MVBench but leads to a marginal drop on Video-MME. This difference likely stems from MVBench containing spatial reasoning tasks, while Video-MME focuses more on event comprehension. To mitigate this trade-off, we further experimented with mixed fine-tuning, combining a small proportion (5% and 10%) of traditional data from ShareGPT4Video [57] with ScanForgeQA. Results show that

---

[4]`https://deepmind.google/technologies/gemini/pro/`.

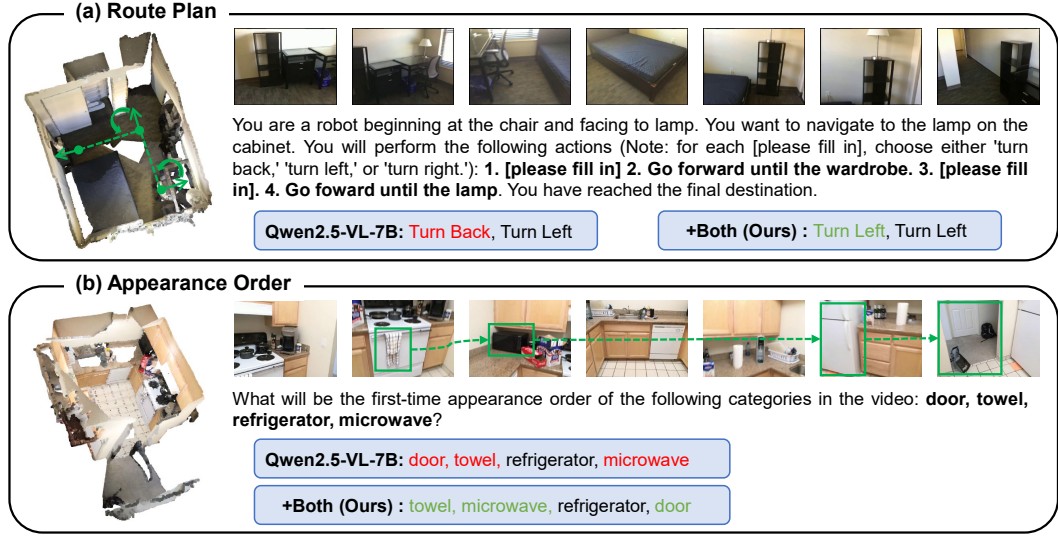

Figure 6: Two examples from VSI-Bench comparing predictions from Qwen2.5-VL-7B and Ours.

this strategy achieves improved performance, surpassing the original Qwen2.5-VL-7B baseline, suggesting that spatial fine-tuning can be harmonized with broader capabilities through data balancing.

## 5.2 Ablation Study

In this section, we explored the impact of various design choices, including prompting strategies, fine-tuning datasets, frame sampling strategies, and input resolution, on the performance of VLMs.

**On prompting strategy.** To isolate the contributions of each component in the SpatialMind prompting strategy, we evaluated two variants: one containing only the question component (CoT-Question) and another containing only the scene description (CoT-Scene). As shown in Table 4, both variants independently improve spatial reasoning performance, but are less effective than the full combined prompt. Notably, the scene description contributes more significantly to model performance than the reasoning steps, suggesting its central role in facilitating spatial understanding.

**On fine-tuning data.** To investigate the effectiveness of our proposed ScanForgeQA against existing spatial datasets, we fine-tuned Qwen2.5-VL-7B on SQA3D [54] and ScanQA [53]. As shown in Table 4, fine-tuning on either of these datasets results in lower performance compared to ScanForgeQA, and even reduces accuracy on tasks involving precise spatial estimation (e.g., *Room Size*). This is primarily due to the limited presence of fine-grained spatial estimation samples in the existing datasets. Importantly, both datasets and the VSI-Bench benchmark originate from the same source (i.e., ScanNet [31]), resulting in minimal data discrepancy. This contrast emphasizes the advantage of our simulated data generation pipeline.

**On frames and resolution.** To evaluate the robustness of our approach, we analyzed performance sensitivity to the number of input frames and image resolution. Figure 5 visualizes the performance of the +Both variant and the baseline Qwen2.5-VL-7B under various configurations. Our method consistently outperforms the baseline across all settings, with performance further improving as the number of frames and resolution increase. This indicates that our approach remains stable and effective under varying visual input conditions.

## 5.3 Qualitative Analysis

In Figure 6, we presented two illustrative examples from VSI-Bench, comparing predictions from the baseline Qwen2.5-VL-7B and our enhanced variant (+Both). In Case (a), Qwen2.5-VL-7B fails to produce the correct directional prediction, likely due to its limited capacity for 3D spatial reasoning. In contrast, our method successfully identifies the correct answer. Case (b) involves a simpler spatial reasoning task, however, Qwen2.5-VL-7B still fails, potentially due to insufficient object localization.

Our enhanced variant, benefiting from both structured prompting and spatially grounded fine-tuning, demonstrates notable improvements in accuracy and reasoning robustness.

# 6 Conclusion

In this work, we present an effective framework for enhancing visual-spatial reasoning in VLMs without modifying their underlying architecture. This makes our approach readily adaptable across models of varying scales and types. By integrating the structured prompting strategy (SpatialMind) with an automatically constructed dataset (ScanForgeQA), we enable VLMs to more effectively interpret and reason about 3D spatial relationships in complex visual scenes. Extensive evaluations across multiple spatial reasoning benchmarks demonstrate that our framework consistently improves accuracy, robustness, and generalization. Furthermore, our analysis reveals that prompting and fine-tuning play complementary roles in advancing visual-spatial understanding.

## Acknowledgments and Disclosure of Funding

This work is supported by Shenzhen Science and Technology Program, No.:KQTD20240729102207002; the National Natural Science Foundation of China, No.:62376140, No.:62476071, No.:625B2065, No.:U24A20328, No.:624B2047, and No.:U23A20315; the Major Key Project of Pengcheng Laboratory, No.:PCL2025A14; GuangDong Basic and Applied Basic Research Foundation, No.:2025A1515011732; the Science and Technology Innovation Program for Distinguished Young Scholars of Shandong Province Higher Education Institutions, No.:2023KJ128; the Special Fund for Taishan Scholar Project of Shandong Province; the Fundamental Research Funds for the Central Universities, No.:HIT.DZJJ.2025048 and No.:HIT.DZJJ.2024041.

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

# A  Experimental Settings

## A.1  Benchmarks

To validate the effectiveness of our SpatialMind strategy and ScanForgeQA data in enhancing the visual-spatial understanding capabilities of VLMs, we conducted evaluations on four comprehensive benchmarks: VSI-Bench [1], OpenEQA [52], ScanQA [53], and SQA3D [54]. The comparisons are shown in Table 5.

Table 5: Comparison of different benchmarks. Note that the statistics for the ScanQA and SQA3D datasets are reported on their respective validation sets.

| Dataset | #Questions | #Scenes | Source | Question Types |
|---------|-----------|---------|--------|----------------|
| VSI-Bench | >5,000 | 288 | ScanNet, ScanNet++, ARKitScenes | Configuration, Measurement Estimation, Spatio-temporal Reasoning |
| OpenEQA | 1,899 | >180 | ScanNet, HM3D | Object Recognition, Attribute Reasoning, Spatial Understanding, Functional Reasoning |
| ScanQA | 9,353 | 71 | ScanNet | Object, Color, Quantity, Location |
| SQA3D | 3,261 | 65 | ScanNet | Spatial Relations, Commonsense, Navigation, Multi-hop Reasoning |

### A.1.1  VSI-Bench

VSI-Bench [1] is a comprehensive evaluation benchmark designed to assess the visual-spatial reasoning capabilities of VLMs in dynamic 3D environments. It comprises over 5,000 high-quality question–answer pairs grounded in 288 diverse indoor video scenes, sourced from real-world 3D reconstruction datasets such as ScanNet [31], ScanNet++ [35], and ARKitScenes [29]. The benchmark defines eight distinct tasks organized into three major categories: **configuration-based tasks** (e.g., object counting, relative direction, and path planning), **measurement estimation tasks** (e.g., object size, absolute distance, room size), and **spatiotemporal reasoning tasks** (e.g., object appearance order). These tasks collectively assess VLMs' abilities to reason about spatial relationships, estimate scale, and understand temporal information. Evaluation is primarily based on accuracy, with tolerance-based error metrics applied to numerical estimation tasks to account for minor deviations. All evaluations follow the official protocols defined in VSI-Bench [1].

### A.1.2  OpenEQA

OpenEQA [52], introduced by Meta AI in 2024, is a benchmark designed to evaluate the spatial understanding and reasoning capabilities of VLMs in real-world indoor environments. The dataset comprises over 1,600 human-annotated question–answer pairs spanning 180+ diverse indoor scenes and supports two primary evaluation tasks. In our experiment, we focused on the **Episodic Memory** (EM-EQA) setting, where models must answer questions based solely on a previously observed egocentric video trajectory, without access to external spatial priors. Questions cover a wide range of reasoning types, including object attributes, spatial relationships, object locations, and functional reasoning. To evaluate model perfromance on open-ended responses, we employed GPT-4o [5] as an automatic evaluator to assess semantic alignment between predicted and ground-truth answers. This enables consistent and reliable scoring across diverse question formats.

### A.1.3  ScanQA

ScanQA [53] is a large-scale benchmark designed to evaluate spatial question answering in real-world indoor environments, grounded in richly annotated 3D scans. Built on top of the ScanNet [31] dataset, it contains over 41,000 human-curated question–answer pairs across 800 RGB-D scenes, with each question grounded in the 3D geometry and semantics of the scene. The dataset covers a broad range of open-ended questions involving object attributes, spatial relations, and scene-level understanding,

---

[5] https://github.com/mbzuai-oryx/Video-ChatGPT/tree/main/quantitative_evaluation.

Table 6: Performance comparison on VSI-Bench. † indicates results on VSI-Bench (tiny) set and LLaVA-NV denotes the LLaVA-NeXT-Video model.

| Method | Obj. Count | Abs. Dist. | Obj. Size | Room Size | Rel. Dist. | Rel. Dir. | Route Plan | Appr. Order | Avg | Δ |
|---|---|---|---|---|---|---|---|---|---|---|
| | | | | Close-source | | | | | | |
| Human Level† | 94.3 | 47.0 | 60.4 | 45.9 | 94.7 | 95.8 | 95.8 | 100.0 | 79.2 | - |
| Gemini-2.0 Flash† | 52.4 | 30.6 | 66.7 | 31.8 | 56.0 | 46.3 | 24.5 | 55.1 | 45.4 | - |
| Gemini-1.5 Pro† | 49.6 | 28.8 | 58.6 | 49.4 | 46.0 | 48.1 | 42.0 | 68.0 | 48.8 | - |
| Gemini-1.5 Pro | 56.2 | 30.9 | 64.1 | 43.6 | 51.3 | 46.3 | 36.0 | 34.6 | 45.4 | - |
| +SpatialMind | 63.9 | 51.8 | 70.2 | 47.3 | 56.3 | 45.9 | 42.6 | 44.3 | 52.8 | ↑ 7.4% |
| GPT-4o | 46.2 | 5.3 | 43.8 | 38.2 | 37.0 | 41.3 | 31.5 | 28.5 | 34.0 | - |
| +SpatialMind | 40.0 | 27.1 | 62.7 | 40.9 | 41.0 | 39.6 | 37.1 | 38.5 | 40.8 | ↑ 6.8% |
| | | | | Open-source | | | | | | |
| InternVL2-8B | 23.1 | 28.7 | 48.2 | 39.8 | 36.7 | 30.7 | 29.9 | 39.6 | 34.6 | - |
| +SpatialMind | 35.8 | 28.9 | 49.7 | 44.4 | 37.2 | 34.8 | 35.1 | 45.5 | 38.9 | ↑ 4.3% |
| +ScanForgeQA | 45.3 | 33.4 | 54.8 | 45.0 | 41.1 | 36.1 | 33.4 | 43.0 | 41.5 | ↑ 6.9% |
| +Both | 47.0 | 32.8 | 53.2 | 46.6 | 39.8 | 36.8 | 37.9 | 47.5 | 42.7 | ↑ 8.1% |
| InternVL2-40B | 34.9 | 26.9 | 46.5 | 31.8 | 42.1 | 32.2 | 34.0 | 39.6 | 36.0 | - |
| +SpatialMind | 36.4 | 30.0 | 49.1 | 41.8 | 43.8 | 36.1 | 35.6 | 50.0 | 40.4 | ↑ 4.4% |
| +ScanForgeQA | 51.0 | 29.2 | 52.7 | 38.1 | 47.2 | 36.4 | 35.9 | 47.6 | 42.3 | ↑ 6.3% |
| +Both | 52.2 | 30.5 | 54.4 | 41.0 | 50.5 | 37.0 | 40.2 | 50.3 | 44.5 | ↑ 8.5% |
| InternVL2.5-8B | 7.0 | 34.1 | 43.0 | 42.4 | 38.0 | 40.1 | 23.2 | 35.9 | 33.0 | - |
| +SpatialMind | 18.7 | 29.6 | 46.5 | 45.2 | 40.3 | 40.9 | 27.8 | 46.6 | 37.0 | ↑ 4.0% |
| +ScanForgeQA | 56.5 | 33.2 | 50.7 | 43.4 | 39.0 | 33.1 | 28.4 | 34.3 | 39.8 | ↑ 6.8% |
| +Both | 52.8 | 30.7 | 53.4 | 44.8 | 41.1 | 38.7 | 29.6 | 47.0 | 42.3 | ↑ 9.3% |
| InternVL2.5-38B | 43.6 | 33.0 | 53.0 | 48.8 | 53.5 | 35.7 | 34.5 | 34.0 | 42.0 | - |
| +SpatialMind | 56.1 | 38.3 | 59.3 | 52.3 | 59.7 | 43.6 | 39.0 | 32.9 | 47.7 | ↑ 5.7% |
| LLaVA-NV-7B | 48.5 | 14.0 | 47.8 | 24.2 | 43.5 | 42.4 | 34.0 | 30.6 | 35.6 | - |
| +SpatialMind | 49.0 | 22.6 | 47.9 | 24.6 | 41.3 | 43.0 | 37.1 | 29.8 | 36.9 | ↑ 1.3% |
| LLaVA-NV-72B | 48.9 | 22.8 | 57.4 | 35.3 | 42.4 | 36.7 | 35.0 | 48.6 | 40.9 | - |
| +SpatialMind | 52.5 | 24.5 | 60.2 | 37.7 | 42.7 | 39.5 | 37.3 | 51.0 | 43.2 | ↑ 2.3% |
| VideoLLaMA3-7B | 36.8 | 22.2 | 34.7 | 24.9 | 44.6 | 41.7 | 36.1 | 28.8 | 33.7 | - |
| +SpatialMind | 38.6 | 23.9 | 36.2 | 36.6 | 42.3 | 40.9 | 35.9 | 33.2 | 36.0 | ↑ 2.3% |
| Qwen2-VL-7B | 39.4 | 25.0 | 25.8 | 43.2 | 32.6 | 30.9 | 27.8 | 32.6 | 32.2 | - |
| +SpatialMind | 43.7 | 29.0 | 29.4 | 46.5 | 33.9 | 32.8 | 32.0 | 31.9 | 34.9 | ↑ 2.7% |
| Qwen2.5-VL-7B | 40.3 | 22.2 | 50.1 | 38.9 | 38.0 | 40.7 | 31.4 | 35.9 | 37.2 | - |
| +SpatialMind | 45.1 | 25.2 | 52.1 | 41.4 | 38.7 | 41.6 | 34.7 | 34.5 | 39.2 | ↑ 2.0% |
| +ScanForgeQA | 53.2 | 30.5 | 56.8 | 44.9 | 42.3 | 44.0 | 37.3 | 37.7 | 43.3 | ↑ 6.1% |
| +Both | 55.0 | 29.5 | 57.3 | 44.0 | 43.5 | 44.3 | 38.3 | 39.2 | 43.9 | ↑ 6.7% |
| Qwen2.5-VL-72B | 37.9 | 28.6 | 57.4 | 49.8 | 45.5 | 38.4 | 20.6 | 35.4 | 39.2 | - |
| +SpatialMind | 42.3 | 32.0 | 61.7 | 53.8 | 48.2 | 43.9 | 30.4 | 39.3 | 44.0 | ↑ 4.8% |
| +ScanForgeQA | 45.2 | 32.7 | 63.3 | 52.4 | 50.1 | 41.7 | 32.8 | 40.2 | 44.8 | ↑ 5.6% |
| +Both | 48.6 | 34.4 | 68.9 | 54.7 | 53.4 | 43.9 | 30.1 | 42.7 | 47.1 | ↑ 7.9% |

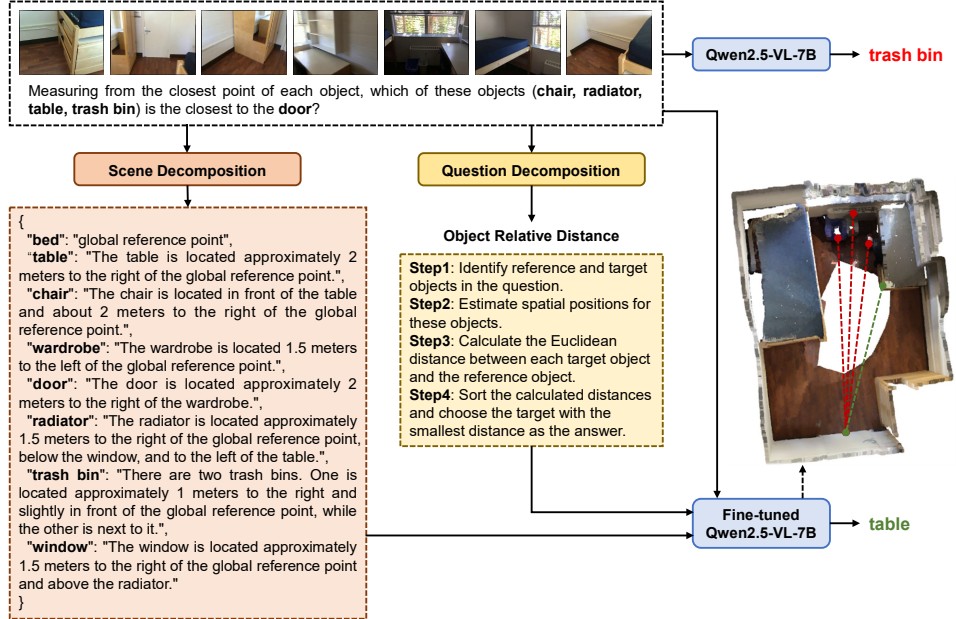

Figure 7: A complete example illustrating the visual prompting process with intermediate outputs.

such as "What is between the sofa and the table?" or "Where is the lamp located?". To evaluate model performance, we adopted BLEU-1 as the primary metric. Given that the answers are typically short and factual, unigram precision offers a reliable measure of response quality.

### A.1.4   SQA3D

SQA3D [54] is a reasoning-centric benchmark designed to evaluate situational understanding within 3D indoor scenes. Built on 650 scenes from ScanNet [31], it comprises approximately 6,800 annotated situations and over 33,000 questions that span a diverse range of reasoning types, including spatial relations, commonsense reasoning, and multi-hop logical reasoning. Each question is grounded in a specific 3D context, requiring precise and spatially informed answers. To evaluate model performance, we used the Exact Match at 1 (EM-1) metric. This metric is well-suited for the benchmark, as answers are typically concise, specific, and context-dependent.

### A.2   Baselines

As shown in Tables 6, our evaluation includes a diverse set of VLM baselines spanning a range of architectures and scales. These include open-source models such as InternVL2 [58] (8B, 40B), InternVL2.5 [58] (8B, 38B), LLaVA-NeXT-Video [59] (7B, 72B), VideoLLaMA3 [60] (7B), Qwen2-VL [61] (7B), and Qwen2.5-VL [62] (7B, 72B). We also evaluated closed-source models, including Gemini-2.0 Flash [6], Gemini-1.5 Pro [7] and GPT-4o [63]. For open-source models, we adopted each model's default parameter settings, including learning rate, number of frames, and input resolution. For closed-source models, GPT-4o processes 16 frames per video, while all Gemini models operate at a fixed sampling rate of 1 frame per second (1 FPS). All experiments are conducted on 8 NVIDIA H20 GPUs.

## B   More Experimental Results

### B.1   Performance Comparison

In Table 6, we extended our evaluation to additional VLMs with varying architectures and parameter scales, applying both the SpatialMind prompting and ScanForgeQA fine-tuning. The results

---

[6] https://deepmind.google/technologies/gemini/flash/.

[7] https://deepmind.google/technologies/gemini/.

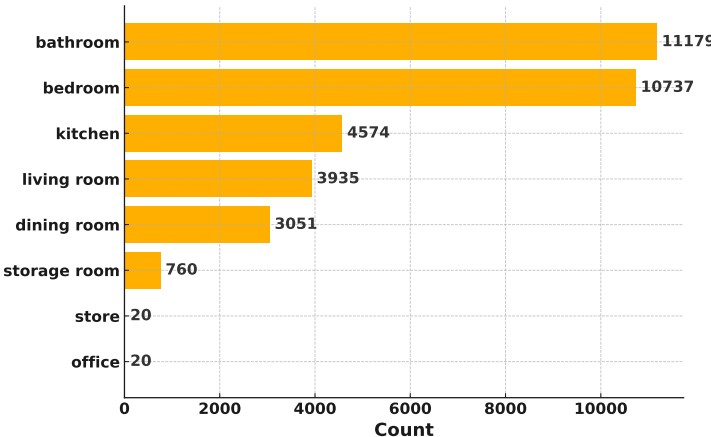

Figure 8: Distribution of room types in the ScanForgeQA dataset.

are consistent with those reported in our main analysis, further reinforcing the effectiveness and generalizability of our proposed methods. Moreover, the combined use of prompting and fine-tuning continues to yield superior performance across models, highlighting their complementary strengths and demonstrating the robustness of our framework.

### B.2 Case Study

Figure 7 presents a detailed example that illustrates the full prompting process along with intermediate outputs. The scene decomposition module produces approximate spatial descriptions of object locations, effectively capturing the overall layout of the 3D scene. Meanwhile, the question decomposition module identifies the question type as "object relative distance" and selects the corresponding reasoning steps to guide inference. By combining this structured information and feeding it into the Qwen2.5-VL-7B model fine-tuned with ScanForgeQA, the correct target object is successfully identified. This example demonstrates the interpretability and effectiveness of our framework in performing multi-step spatial reasoning grounded in visual context.

## C  More details for ScanForgeQA

### C.1 Scene Distribution

Figure 8 presents the distribution of the eight room categories included in ScanForgeQA. The dataset exhibits a clear long-tailed distribution in which categories such as *bathroom* and *bedroom* are heavily represented, whereas *store* and *office* types are comparatively rare. This imbalance is primarily due to the inherent distribution of room types in the 3D-FRONT dataset [47].

### C.2 Scan Example

We provided two frames captured by the camera during the creation of the scanning video, as shown in Figure 9, to visually illustrate this process. Additionally, we further provided scanning creation scripts as well as more video demos in the Supplemental Material:

- "**codes/nav_script.py**" is the script that creates the navigation scan, presenting the exact implementation.
- "**Creation of Scanning Video.mp4**" illustrates the creation process of the scanning video from both first-person and third-person perspectives.
- "**Scanning Video.mkv**" presents the rendered scene scan resulting from the aforementioned scanning process.

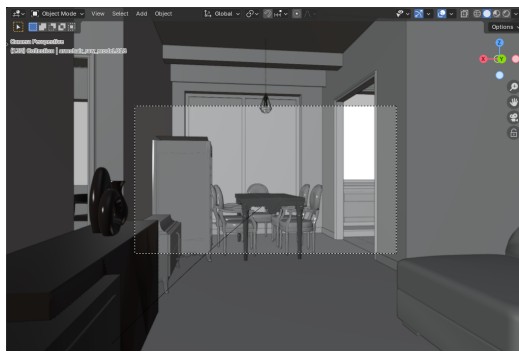 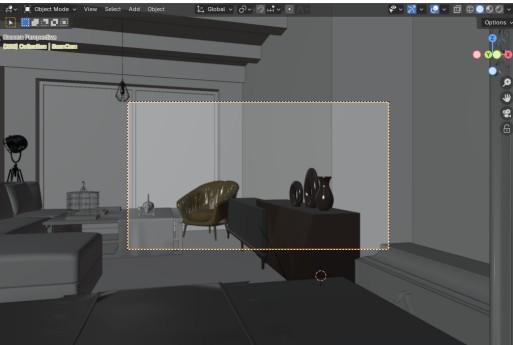

Figure 9: Screenshots from the scan creation process.

### C.3   QA Definition

Table 7 provides representative templates and corresponding answer sources for each question type in ScanForgeQA. This information clarifies the coverage of different reasoning categories and indicates which scene annotations are used to derive ground-truth answers.

## D   More details for SpatialMind

### D.1   Prompt for Object Description

As a representative example, Figure 10 illustrates the prompt used to generate textual descriptions of object positions. This prompt guides the model to produce structured spatial layouts in natural language based on the visual input. Similar prompts are used for other reasoning components, following a consistent design pattern. Other prompts are provided in the "**codes/gen_scene_exp.py**" file of Supplementary Material for reference and reproducibility.

### D.2   Reasoning Steps for Different Questions

Figure 7 provides an example of the detailed reasoning steps used for the *relative distance* question type. Each question type is paired with a concise, structured reasoning process that offers a generalizable solution strategy for VLMs. All reasoning steps for the various question types are included in the "**codes/reason_steps.py**" file of Supplementary Material for reference and reproducibility.

## E   Limitations

This paper demonstrates that existing VLMs tend to rely more heavily on understanding textual descriptions when performing spatial reasoning. However, textual descriptions alone may not capture spatial semantics as intuitively as visual input, potentially limiting the upper bound of performance for such methods. Moreover, we primarily focus on basic single-room 3D scene. Although strong performance has been achieved on OpenEQA [52], which includes some multi-room scenes, more complex settings, such as large-scale indoor layouts and outdoor environments, remain underexplored and represent important directions for future work.

## F   Broader Impacts

Our approach improves visual-spatial understanding in VLMs, enabling better interpretation of object layouts and spatial relationships. This enhancement offers potential benefits in areas such as embodied AI and AR/VR applications, where spatial reasoning is essential. However, similar to other prompting and fine-tuning methods, our framework may inherit biases from pre-trained models or training data, potentially leading to unfair or inaccurate spatial interpretations. While our method reduces reliance on expensive 3D sensors, evaluating spatial accuracy remains a key challenge, especially for

downstream tasks that demand high robustness and trustworthiness. Despite these limitations, we believe that releasing our framework and dataset will foster transparency, support reproducibility, and accelerate progress toward trustworthy and robust spatial reasoning systems.

Table 7: Overview of question templates and their corresponding answer sources.

| Type | Question Template | Answer Source |
|---|---|---|
| **Attribute Estimation** | | |
| Object Count | How many <A> are there in the room? 
 What is the total number of <A>? | Name of object instance |
| Object Size | What is the length of the longest side of <A> in meters? 
 What is the size of <A> in square meters? 
 What is the length of the shortest side of <A> in meters? 
 How tall is <A> in meters? | Dimension of the object model |
| Room Size | What is the size of the room in square meters? | Scene boundaries and scale |
| Room Type | Based on object layout, what is the most likely type of this room? 
 Is this space a living room, a kitchen, or something else? | Room labels |
| **Spatial Reasoning** | | |
| Relative Distance | Which of these objects (<A>, , <C>) is the closest to <R>? 
 Among the listed objects (<A>, , <C>), which one is closest to <R>? | The minimum distance between the centroids of objects |
| Absolute Distance | What is the distance between <A> and  in meters? 
 Measure the distance from <A> to  in meters. 
 How far is <A> from  in meters? | Euclidean distance between objects |
| Relative Direction | If I am standing by <A> and facing , which side is object <R> on? 
 From the viewpoint at <A> facing , where is <R>? | The direction of connections between objects |
| Contact Relationship | Is there a gap between <A> and ? 
 Are <A> and  touching each other? | Comparison of distances and sizes between objects |
| **Hypothesis Analysis** | | |
| Operation Feasibility | Considering only object sizes, is there enough space to put <A> in ? 
 Considering only object dimensions, is it feasible to place <A> on ? | Comparison of sizes between objects |

[Task]
You are given a video (multiple frames) capturing an indoor scene. Your goal is to recognize {categories_of_interest} objects, analyze the spatial layout of the scene, and describe the relative position of each object.

[Instructions]
1. Per-frame analysis:
- For each frame, choose one object as a **local reference point**.
- Predict the relative position of all other objects with respect to this local reference point.
- Express relative positions using simple terms like "left", "right", "front", "behind", "above", "below", and approximate distances (e.g., "2 meters to the right").

2. Global scene layout:
- Take the **local reference point from the first frame** as the **global reference point** for the whole video.
- Use overlapping objects between frames to align frames together.
- Gradually build the spatial descriptions for all objects relative to the global reference point.

[Rules]
- If a category has multiple instances (e.g., two chairs), describe each instance separately.
- Preserve the real-world spatial relationships and distances as accurately as possible.
- Use clear and consistent directional and distance terms.

[Output Format]
ONLY Return the result as a JSON dictionary following STRICTLY this format:
{
"category name": "global reference point",
"another category name": "Position description relative to the global reference point",
...
}

Example:
{
"chair": "The chair is located 1.5 meters to the left and 0.5 meters behind the global reference point",
"table": "The table is located 2 meters to the right of the global reference point",
"lamp": "The lamp is located 1 meter above and 1 meter behind the global reference point"
}

Figure 10: Example prompt used to generate textual descriptions of object positions.

