# OpenReview forum: "Spatial Understanding from Videos: Structured Prompts Meet Simulation Data"
_NeurIPS.cc/2025/Conference — NeurIPS 2025 spotlight_

### Official Review · Reviewer_qZj9 · 2025-06-06

**Clarity:** 3
**Significance:** 3
**Originality:** 3
**Rating:** 5
**Confidence:** 5

**Summary:**

This paper presents a general framework for enhancing the VLMs’ spatial understanding ability, consisting of two parts:
-	SpatialMind: A structured chain-of-thought prompting strategy that divides the spatial understanding problems into multiple steps for reasoning.
-	ScanForgeQA: A pipeline for automatically constructing spatial understanding QA data in the simulation environments is proposed, and a large-scale QA dataset for spatial reasoning is constructed based on this pipeline.
The experimental results show that both parts can effectively improve the performance of various VLMs on multiple benchmarks. Moreover, combining the two parts can further enhance the performance.

**Questions:**

1.	What are the specific experimental settings in the comparative experiments of different representation formats, and what are the detailed settings for fine-tuning on ScanForgeQA dataset?
2.	The SpatialMind pipeline relies on GPT-4o for both scene decomposition and question decomposition. How it works if totally use the evaluated VLM in the pipeline?
3.	Concerns about VLM’s hallucination: During scene decomposition process, will there be a situation where the local coordinates estimated by VLM for each frame conflict with each other? If such a situation exists, then how can they be integrated into global coordinates?

**Ethical Concerns:**

["NO or VERY MINOR ethics concerns only"]

**Final Justification:**

I recommend the paper to be accepted.

**Limitations:**

Yes.

**Quality:**

3

**Strengths And Weaknesses:**

# Strengths
-	This paper is well-structed and the detailed information of the method is provided
-	The proposed SpatialMind strategy and ScanForgeQA dataset is easy to apply and applicable to various VLMs, thus it’s valuable for future research
-	The paper investigates which type of scene representation is more effective for VLMs.

# Weaknesses
-	Will the accuracy of coordination estimation affect the model performance? The authors should explain more about it.

-      What is the gap between synthesis data and real data?

---

> ### Author Rebuttal · Authors · 2025-07-31
>
> We appreciate the reviewer’s insightful comments and helpful suggestions. Below are our responses and explanations:
>
> > **W1**: Will the accuracy of coordination estimation affect the model performance? The authors should explain more about it.
>
> **A1**：The accuracy of coordinate estimation has a direct and significant impact on the overall performance of models in spatial reasoning tasks. Specifically, the estimated spatial coordinates of objects are provided to VLMs alongside the scanned video for answer prediction, meaning that model decisions are influenced by the estimated coordinates. If these estimates are inaccurate, for example if the predicted location of a target object deviates from its actual position, subsequent spatial reasoning (such as determining whether something is to the "left", "right", "front", or "behind") may be misled, resulting in incorrect answers.
>
> In our experiments (as shown in Figure 3 of the manuscript), we observed that more detailed coordinate representations (e.g., 3D map) yield less improvement than coarser textual position descriptions. This is mainly because current VLMs are not yet capable of accurately interpreting fine-grained spatial coordinates, and more detailed representations may introduce additional noise (or hallucination). Therefore, improving the accuracy of coordinate estimation is essential for further enhancing a model's spatial reasoning capabilities.
>
> > **W2**: What is the gap between synthesis data and real data?
>
> **A2**: Regarding the gap between synthetic data and real data, we summarize the following four aspects:
>
> | Comparison Aspect | Synthetic Data  | Real Data |
> |-------------------------------|---------------------------------------------------------------------|--------------------------------------------------------------------|
> | Scene Diversity               | Covers various room types (living room, bedroom, kitchen, office, etc.), offering rich diversity | Usually limited to a few scene types (mostly kitchens and daily environments), with less diversity |
> | Layout Flexibility | Furniture types, quantities, and positions can be flexibly adjusted; scenes can be quickly generated or modified programmatically with low cost | Once scanned, layout and objects are fixed; modification and expansion are difficult and costly    |
> | Data Scale                    | Easy to generate large-scale, diverse datasets in bulk              | Data scale is limited by real-world collection and difficult to expand on a large scale            |
> | Realism                       | Visual and physical properties can be realistically simulated, but may still differ from real-world scenarios | Directly collected from the real world, providing high authenticity                                |
>
> As shown in the table, synthetic data can be generated at scale, supports diverse scene types, and allows flexible modifications to layouts. However, it may lack the full authenticity of real-world environments. In contrast, real data offers high fidelity and realism but is constrained by limited scene diversity, difficulty in modification, and challenges in scalability.
>
> > **Q1**: What are the specific experimental settings in the comparative experiments of different representation formats, and what are the detailed settings for fine-tuning on ScanForgeQA dataset?
>
> **A3**: For the experiments involving different representation formats (as shown in Figure 3 of the manuscript), we performed inference using the default evaluation code provided by VSI-Bench, which is built on top of the lmms_eval framework. The experimental settings are as follows. For all other parameters not explicitly stated, we followed the default configurations.
>
> | Parameter|GPT-4o | Qwen2.5-VL |
> |------|------|------|
> | Version | 2024-08-06 | 7B/72B |
> | Frames | 16 | 16 |
>
> For the fine-tuning experiments on the ScanForgeQA dataset, we used the official codebase provided by InternVL2. For the Qwen2.5-VL model, fine-tuning was conducted using the LLaMA-Factory framework. The detailed settings are as follows:
>
> | Parameter           | InternVL2 (8B) | InternVL2 (40B) | Qwen2.5-VL (7B) | Qwen2.5-VL (72B) |
> |---------------------|----------------|-----------------|-----------------|------------------|
> | Frames              | 8              | 8               | 16              | 16               |
> | Batch Size          | 32             | 16              | 32              | 8                |
> | Learning Rate       | 4e-5           | 2e-5            | 2e-5            | 1e-5             |
> | Epochs              | 1              | 1               | 1               | 1                |
> | Optimizer           | AdamW          | AdamW           | AdamW           | AdamW            |
> | Image Resolution    | 224×224        | 224×224         | 224×224         | 224×224          |
> | Max Sequence Length          | 4096           | 4096            | 2048            | 2048             |
> | Weight Decay        | 0.05           | 0.05            | 0.01               | 0.01                |
> | Warmup Ratio        | 0.03           | 0.03            | 0.03            | 0.03             |
> | LoRA Rank           | 8              | 8               | 8               | 8                |
> | LoRA Alpha          | 16             | 16              | 16              | 16               |
>
> All experiments are conducted on 8 NVIDIA H20 GPUs. We will include the detailed experimental settings mentioned above in the supplementary materials and provide the full implementation details in the open-source repository.
>
> > **Q2**: The SpatialMind pipeline relies on GPT-4o for both scene decomposition and question decomposition. How it works if totally use the evaluated VLM in the pipeline?
>
> **A4**: In the scene decomposition of the SpatialMind pipeline, we employ GPT-4o to extract all objects mentioned across questions related to the same scene, and then prompt the VLMs (e.g., Qwen2.5-VL) to carry out the subsequent reasoning process. GPT-4o is selected for its high API concurrency efficiency, low preprocessing latency, and strong performance. Nevertheless, the entire pipeline can also be executed using the evaluation VLM itself, simply by replacing GPT-4o with the target model in this step.
>
> To investigate the potential impact of this modification, we conducted an experiment on VSI-Bench using the Qwen2.5-VL-7B model. The results are shown in the table below.
>
> | Method | Average Accuracy |
> |------|------|
> | Qwen2.5-VL-7B | 37.2|
> | +SpatialMind (Original) | 39.2 |
> | +SpatialMind (w/o GPT-4o) | 38.6 |
>
> The results show a decline in performance after the modification. Given that extracting objects from the question during scene decomposition is a relatively simple task with low potential for error, we speculate that the performance drop may stem from inaccuracies in the reasoning steps generated by the VLMs during question decomposition.
>
> > **Q3**: Concerns about VLM’s hallucination: During scene decomposition process, will there be a situation where the local coordinates estimated by VLM for each frame conflict with each other? If such a situation exists, then how can they be integrated into global coordinates?
>
> **A5**: Yes, hallucination is a common issue in VLMs, which can lead to conflicts or inconsistencies in local coordinate estimations across different frames. This phenomenon may explain why the generated 3D map and 2D grid perform worse than the text-based position descriptions in Figure 3 of the manuscript. As the granularity of the generated representation increases, the likelihood and severity of hallucinations also increase. In our experiments, we did not focus on addressing this issue; instead, we chose to use text-based position descriptions generated by VLMs as the scene representation format, as they yielded better overall performance.

---

> > ### Comment · Reviewer_qZj9 · 2025-08-01
> >
> > I would like to sincerely thank the authors for the response. My concerns are largely addressed and I think the paper is good to be accepted to NeurIPS.

---

> > > ### Author Response · Authors · 2025-08-03
> > >
> > > We sincerely thank the reviewer for the thoughtful and constructive feedback. We are pleased that our responses have largely addressed your concerns, and we deeply appreciate your kind support and recommendation for acceptance.

---

### Official Review · Reviewer_mMuG · 2025-07-03

**Clarity:** 3
**Significance:** 4
**Originality:** 3
**Rating:** 5
**Confidence:** 4

**Summary:**

This paper aims to address the limitation of state-of-the-art vision-language models (VLMs) in performing spatial reasoning over 3D scenes. Specifically, it introduces the SpatialMind approach, which decomposes 3D spatial reasoning into structured multi-step reasoning for complex scenes and questions. Furthermore, to train the approach, the authors also propose the large-scale ScanForgeQA dataset, which includes questions on attribute estimation, spatial reasoning, and hypothesis analysis to encourage fine-grained spatial learning. The paper demonstrates the effectiveness of the proposed SpatialMind approach by showing consistent improvements in spatial reasoning performance across the VSI-Bench, OpenEQA, ScanQA, and SQA3D benchmarks.

**Questions:**

Please look at the abovementioned weaknesses. It will be much more insightful to include empirical comparisons to more VLMs that are targeted at spatial reasoning.

**Ethical Concerns:**

["NO or VERY MINOR ethics concerns only"]

**Final Justification:**

The authors have addressed my concerns.

**Limitations:**

Yes

**Quality:**

3

**Strengths And Weaknesses:**

Strength 1 - In terms of clarity and quality of the paper, the paper is relatively well-written and motivated. The model figures are informative and especially helpful in helping the reader to understand the proposed SpatialMind  approach as well as the data generation pipeline.

Strength 2 - The contributions of this work are significant since it addresses an important research problem. In particular, the data generation pipeline introduced to curate ScanForgeQA provides a cost-effective way to automatically generate large-scale, high-quality QA datasets using 3D simulation environments and LLM-guided scene synthesis. Both dataset and pipeline can be useful to further research in this topic.

Strength 3 - Due to the way that the SpatialMind approach incorporates scene and question decomposition into step-wise reasoning, it can be beneficial for generating rationales that are more interpretable.

Weakness 1 - The empirical results included in the paper do not compare to some VLMS that are specifically targeted at 3D spatial reasoning and understanding, such as 3D-LLM and 3DLLM-Mem among many others. It will be much more insightful if such empirical comparisons are included.

---

> ### Author Rebuttal · Authors · 2025-07-31
>
> We appreciate the reviewer’s insightful comments and helpful suggestions. Below are our responses and explanations:
>
> > **W1**: The empirical results included in the paper do not compare to some VLMS that are specifically targeted at 3D spatial reasoning and understanding, such as 3D-LLM and 3DLLM-Mem among many others. It will be much more insightful if such empirical comparisons are included.
>
> **A1**: To enable a more comprehensive comparison, we included two additional categories of baselines focused on spatial understanding. The first category uses 3D point clouds as input, such as 3D-LLM and Chat-Scene. The second category consists of concurrent works on arXiv that also take scanned videos as input, including Struct2D, SpaceR-7B, and Video-R1-7B. The experimental results are presented in the table below, with corresponding explanations provided as follows.
>
> | Method | Video-Input Only | ScanQA Val. (BLEU-1) | VSI-Bench (Avg. Accuracy)|
> |------|------|------|------|
> | 3D-LLM (BLIP2-FlanT5) | ✖ | 39.3 | -|
> | Chat-Scene | ✖ | 43.2 |-|
> | Struct2D | ✔ | - | 41.9 |
> | SpaceR-7B | ✔ | - | 41.6 |
> | Video-R1-7B | ✔ | - | 34.6 |
> | Qwen2.5-VL-7B | ✔ | 32.5 | 37.2 |
> | +Both (Ours) | ✔ | 37.9 | **43.9** |
> | Qwen2.5-VL-72B | ✔ | 35.4 | 39.2 |
> | +Both (Ours) | ✔ | **44.1** | **47.1** |
>
> Explanation:
> 1. Since VSI-Bench does not provide point cloud data, methods based on 3D point clouds (e.g., 3D-LLM and Chat-Scene) are unable to produce corresponding results.
> 2. Baseline 3DLLM-Mem has not reported experimental results on either of the two datasets mentioned above.
> 3. The concurrent works that take scanned videos as input (e.g., Struct2D, SpaceR-7B, and Video-R1-7B) have not reported results on the ScanQA dataset.
>
> Based on the experimental results shown in the table, we draw the following observations:
> 1. Compared to spatial understanding baselines that use 3D point clouds as input, our method achieves competitive performance using only 2D scanned video as input.
> 2. In comparison with concurrent works that also take scanned videos as input, our framework demonstrates a significant performance advantage.
> 3. Our method is highly flexible and can substantially enhance spatial reasoning capabilities across different backbone models.
>
> For completeness, we will include the above results and analysis, and cite all relevant references (including 3DLLM-Mem) in the revised manuscript.

---

> > ### Author Response · Authors · 2025-08-06
> >
> > Dear Reviewer mMuG,
> >
> > We sincerely appreciate your recognition and support of our work, as well as the valuable suggestions you provided. We would like to follow up to see if our responses have helped address your questions. If there is anything further we can elaborate on during the discussion period, we would be glad to do so.

---

> > ### Comment · Reviewer_mMuG · 2025-08-07
> >
> > Thank you for the detailed responses. My concerns are addressed and I agree with the strengths raised by the other reviewers.

---

> > > ### Author Response · Authors · 2025-08-07
> > >
> > > Thank you for your kind recognition. We are very pleased that our response has addressed your concerns. Should you require any further clarification during the discussion phase, please do not hesitate to let us know.

---

### Official Review · Reviewer_9tnn · 2025-07-03

**Clarity:** 3
**Significance:** 3
**Originality:** 3
**Rating:** 4
**Confidence:** 3

**Summary:**

This paper identifies key challenges faced by current video models in spatial reasoning tasks and proposes the SpatialMind prompting strategy, which enhances reasoning by first constructing a cognition map. In addition, the authors introduce ScanForgeQA, a large-scale 3D scene reasoning dataset, and provide detailed descriptions of its generation pipeline. Experimental results demonstrate the effectiveness of the proposed SpatialMind approach and ScanForgeQA data.

**Questions:**

- In the "both" setting, is the ScanForgeQA fine-tuning data also annotated using the SpatialMind prompting strategy?
- How does SpatialMind prompting affect inference time?
- Are there any observed cases where SpatialMind prompting misleads the model, leading it to predict incorrect answers?
- What are the authors' thoughts on the effectiveness of general video pretraining for spatial reasoning tasks? For instance, in Table 1, if ScanForgeQA were replaced with a similarly sized dataset of general video QA, would performance still improve?

**Ethical Concerns:**

["NO or VERY MINOR ethics concerns only"]

**Final Justification:**

The two strengths: 1. discussing the connection between spatial reasoning and video training. 2. curating a complicated dataset. These two contributions both are valuable to the spatial reasoning research community. Therefore, I choose my final rating of borderline accept.

**Limitations:**

Yes

**Quality:**

3

**Strengths And Weaknesses:**

### Strengths
- The paper clearly poses the connection between spatial reasoning and video training, a relationship that remains largely underexplored in current VLM research.
- Developing a complicated dataset contributes valuable resources to the spatial reasoning research community.
### Weaknesses
- In Figure 4, the comparison involves training with ScanForgeQA and a small portion of ShareGPT4Video. For a more comprehensive ablation, it would be helpful to include results from fine-tuning on ShareGPT4Video data alone.
- The performance results on ScanForgeQA should be reported explicitly.

---

> ### Author Rebuttal · Authors · 2025-07-31
>
> We appreciate the reviewer’s insightful comments and helpful suggestions. Below are our responses and explanations:
>
> > **W1**: In Figure 4, the comparison involves training with ScanForgeQA and a small portion of ShareGPT4Video. For a more comprehensive ablation, it would be helpful to include results from fine-tuning on ShareGPT4Video data alone.
>
> **A1**: Following your suggestion, we have fine-tuned Qwen2.5-VL-7B model using only the ShareGPT4Video data. The experimental results on the MVBench and Video-MME benchmarks can be found in the sixth row of the table below.
>
> | Fine-tuned data | MVBench  | Video-MME (w sub) |
> |------|------|------|
> | Base | 69.6 | 71.6 |
> | +ScanForgeQA | 69.8 | 70.7 |
> | +ScanForgeQA & 5%ShareGPT4Video| 69.9 | 71.5 |
> | +ScanForgeQA & 10%ShareGPT4Video| 69.9 | 71.8 |
> | +ShareGPT4Video | 69.7 | 72.0 |
>
> The results indicate that for MVBench, which includes a subset of spatial reasoning tasks, incorporating ScanForgeQA yields better performance than ShareGPT4Video. However, neither dataset alone surpasses the performance achieved by combining both. In contrast, for Video-MME, which is designed to evaluate event understanding, the general-purpose ShareGPT4Video dataset is more closely aligned with the evaluation tasks and therefore achieves the best performance.
>
> > **W2**: The performance results on ScanForgeQA should be reported explicitly.
>
> **A2**: We apologize if we have misunderstood your intent, and we appreciate the opportunity to clarify. Our preliminary response is as follows:
>
> ScanForgeQA is a fine-tuning dataset that we constructed, which does not include validation or test sets, and therefore, we do not provide evaluation results on this dataset. The fine-tuning results of various model architectures using this dataset can be found in Table 1 of the manuscript (rows marked with "+ScanForgeQA"). If you could kindly provide further clarification regarding this question, we would be more than happy to offer a more detailed and specific response.
>
> > **Q1**: In the "both" setting, is the ScanForgeQA fine-tuning data also annotated using the SpatialMind prompting strategy?
>
> **A3**: No, the ScanForgeQA fine-tuning data is not annotated using the prompting strategy. In the "both" setting, we first fine-tune the VLMs using the ScanForgeQA data, and then apply the SpatialMind prompting strategy for inference with the fine-tuned VLMs.
>
> > **Q2**: How does SpatialMind prompting affect inference time?
>
> **A4**: The impact of the SpatialMind prompting strategy on inference time is primarily due to the increased prompt length. To evaluate the effect of this strategy on inference speed, we conducted experiments using the Qwen2.5-VL-7B model on an A100 GPU. Specifically, we performed inference on 100 samples and report the average inference time per sample. The results are shown in the table below.
>
> | Method | Time | Time (with Flash Attention) |
> |------|------|------|
> | Base | 8.37s | 5.71s |
> | +SpatialMind | 10.84s | 6.03s|
>
> As observed, applying the SpatialMind prompting strategy does lead to slower inference, which is attributable to the increased number of input tokens. However, this impact can be mitigated through inference acceleration techniques for VLMs such as Flash Attention. Moreover, when dealing with large-scale data, methods such as vLLM can be further employed to narrow the gap between the two.
>
> > **Q3**: Are there any observed cases where SpatialMind prompting misleads the model, leading it to predict incorrect answers?
>
> **A5**: Yes. Although the SpatialMind prompting strategy demonstrates outstanding performance in enhancing spatial reasoning abilities, there are indeed rare cases where the prompts may inadvertently mislead the model, resulting in incorrect predictions. This is mainly due to inaccurate object recognition or significant errors in distance estimation, which can cause intermediate mistakes to accumulate and affect the final outcome. Such issues are common among many prompting methods and can be alleviated by controlling the intermediate results. Despite these isolated cases, our experiments show that SpatialMind prompting is generally effective at decomposing tasks and guiding model reasoning, leading to a significant reduction in systematic errors.
>
> > **Q4**: What are the authors' thoughts on the effectiveness of general video pretraining for spatial reasoning tasks? For instance, in Table 1, if ScanForgeQA were replaced with a similarly sized dataset of general video QA, would performance still improve?
>
> **A6**: We argue that current mainstream general video pretraining datasets primarily focus on the semantic understanding of temporal events in videos, such as action recognition and event relationships, rather than on fine-grained 3D spatial perception and reasoning. Consequently, models pretrained on such datasets struggle to transfer effectively to 3D spatial reasoning tasks.
>
> To further validate this, we conducted an experiment in which we replaced our spatial dataset, ScanForgeQA (containing 925K samples), with a similarly sized general video QA dataset, HowToVQA69M (925K samples randomly selected). The experiments were carried out using the Qwen2.5-VL-7B model, and the results on VSI-Bench are presented in the table below.
>
> | Method | Object Count | Relative Distance | Route Plan | Avg |
> |------|------|------|------|------|
> | Base | 40.3 | 38.0 | 31.4 | 37.2 |
> | +ScanForgeQA | **53.2** | **42.3** | **37.3** | **43.3** |
> | +HowToVQA69M (925K) | 40.7 | 35.9 | 29.8 | 35.6 |
>
> We observe that this replacement does not improve performance on spatial reasoning tasks; instead, it leads to a performance drop due to differences in the data. This further highlights the limitations of general video QA datasets in effectively supporting 3D spatial reasoning. Therefore, we believe that dedicated spatial datasets are both essential and irreplaceable for enhancing a model’s spatial reasoning capabilities.

---

> > ### Comment · Reviewer_9tnn · 2025-08-04
> >
> > Thanks for your clarifications. Most of my questions have been resolved. I will continue to follow the discussions and feedback from other reviewers.

---

> > > ### Author Response · Authors · 2025-08-06
> > >
> > > We greatly appreciate the reviewer’s insightful and constructive comments. We are glad that our responses have addressed most of your questions. We are happy to provide any further clarification if needed.

---

### Official Review · Reviewer_YFSW · 2025-07-03

**Clarity:** 3
**Significance:** 3
**Originality:** 2
**Rating:** 5
**Confidence:** 4

**Summary:**

- This paper presents a unified framework for enhancing 3D spatial reasoning in pre-trained VLMs without modifying their architecture. Specifically, the paper introduces SpatialMind, a structured Chain-of-Thought (CoT) prompting strategy, and ScanForgeQA, a large-scale synthetic QA dataset generated from diverse 3D simulation scenes.
- Experimental results validate the effectiveness of both SpatialMind and ScanForgeQA.

**Questions:**

Regarding the ScanForgeQA dataset, are there examples of questions whose answers cannot be directly inferred from metadata alone?

**Ethical Concerns:**

["NO or VERY MINOR ethics concerns only"]

**Final Justification:**

The paper proposes a scalable dataset generation pipeline, along with ScanForgeQA, a synthetic spatial question-answering dataset that could serve as a valuable resource for the research community. During the rebuttal, the authors also provide comparison with existing 3D QA datasets. Overall, I believe the paper is good to be accepted.

**Limitations:**

yes

**Quality:**

2

**Strengths And Weaknesses:**

**Strengths**
- The paper introduces SpatialMind, a novel structured CoT prompting strategy that enables step-by-step spatial reasoning in VLMs, improving interpretability and performance.
- The paper proposes ScanForgeQA, a large-scale synthetic QA dataset using an automated scene construction pipeline.
- The paper has evaluated the effectiveness of proposed methods on various VLMs.

**Weaknesses**
- The paper proposes SpatialMind, a structured Chain-of-Thought (CoT) prompting strategy, but it lacks comparison with existing baseline prompting strategies.
- The paper would benefit from a comparative analysis between ScanForgeQA and other existing 3D QA datasets such as SPARTUN3D [1], MSQA [2], and 3D-LLM [3], etc. to highlight the novelty and advantages of ScanForgeQA.
- It would be better if the paper could provide some sample instances from the ScanForgeQA dataset

[1] SPARTUN3D: Situated Spatial Understanding of 3D World in Large Language Models

[2] Multi-modal Situated Reasoning in 3D Scenes

[3] 3D-LLM: Injecting the 3D World into Large Language Models

---

> ### Author Rebuttal · Authors · 2025-07-29
>
> We appreciate the reviewer’s insightful comments and helpful suggestions. Below are our responses and explanations:
>
> > **W1**: The paper proposes SpatialMind, a structured Chain-of-Thought (CoT) prompting strategy, but it lacks comparison with existing baseline prompting strategies.
>
> **A1**: To establish comparative baselines, we utilized both GPT-4o (which we used to develop our SpatialMind) and GPT-4.1 (a more advanced model) to generate two distinct prompts focused on spatial understanding, as detailed below.
>
> ```
> # Baseline prompt 1 from GPT-4o
>
> You are a spatial reasoning expert. The input is a scanned indoor video. Your task is to analyze the video to extract accurate spatial information and use it to answer specific questions about the scene.
>
> Step 1: Scene Understanding
> From the video, identify:
> -The room structure: walls, doors, windows
> -Objects: such as a refrigerator, microwave, window, sofa, TV, etc.
> -Estimate the location of each object, using relative coordinates or spatial descriptions (e.g., "near the left wall", "close to the door", etc.)
>
> Step 2: Spatial Reasoning and Question Answering
> -Use the extracted spatial information to answer the given question.
> -You may refer to their positions using coordinates, directional terms (e.g., left, right, front), or visual layout based on the video.
>
> Assume that you have 3D spatial awareness and can mentally reconstruct the layout of the room from the video.
> ```
>
> ```
> # Baseline prompt 2 from GPT-4.1
>
> Carefully watch the indoor scan video and follow these steps:
> 1.Identify and list all relevant objects and features visible.
> 2.Locate each object in the room relative to fixed points (like doors or corners).
> 3.Map the spatial layout of the room, marking object positions.
> 4.Analyze the specific spatial question using visual cues (distance, direction, or size).
> 5.Explain your reasoning step by step, citing evidence from the video.
> 6.Provide a clear, justified answer.
>
> If you are uncertain due to missing or unclear information, mention it and specify what would resolve it.
> ```
>
> We then evaluated the performance of different prompts on the VSI-Bench benchmark using both configurations of the Qwen2.5-VL model, as presented in the table below.
>
> | Source  | Prompt  | Avg (7B) | Avg (72B) |
> |------|------|------|------|
> | VSI-Bench (Default) | Answer with the option's letter from the given choices directly | 37.2 | 39.2 |
> | GPT-4o | Above prompt 1 | 37.6 | 41.9 |
> | GPT-4.1 | Above prompt 2 | 37.5 | 42.3 |
> | Ours | SpatialMind | **39.2** | **44.0** |
>
> The results in the table indicate that different prompting strategies have a substantial impact on the spatial reasoning abilities of VLMs, and our SpatialMind prompting strategy proves more effective in enhancing spatial understanding. We will include the above comparison in the supplementary materials.
>
> > **W2**: The paper would benefit from a comparative analysis between ScanForgeQA and other existing 3D QA datasets such as SPARTUN3D, MSQA, and 3D-LLM, etc. to highlight the novelty and advantages of ScanForgeQA.
>
> **A2**: Following your suggestion, we compared SPARTUN3D, MSQA, 3D-LLM, and our proposed ScanForgeQA dataset from multiple aspects, as shown in the table below.
>
> | Dataset  | Data Source | Scene Number | Scene Fromat | Question Type | QA Number | Scalability |
> |------|------|------|------|------|------|------|
> | SPARTUN3D | 3RScan | 478|Point Cloud | 4 | 133K | Difficult |
> | MSQA | ScanNet、3RScan、ARKitScenes | 1.7K|Point Cloud | 9 | 251K | Difficult |
> | 3D-LLM | ScanNet、HM3D | 1.2K|Point Cloud | 9 | 300K | Difficult |
> | ScanForgeQA (Ours) | Simulation/Synthesis | 34K|Scan Video | 9 |925K | Easy |
>
> We adopted a novel simulated data generation pipeline, which brings the following advantages:
>
> - **Diverse scenes**. Existing datasets all utilize fixed real-world scene data, resulting in limited and homogeneous scene types. In contrast, synthetic scenes feature more diverse furniture and spatial layouts, leading to a richer and more comprehensive variety of scene types.
> - **Large scale**. The number of QA pairs is substantially greater, ranging from three to seven times more than those found in the other datasets.
> - **Easily extensible**. The dataset is easily scalable, as scene compositions can be modified by adjusting furniture layouts or changing generation prompts. This level of flexibility is difficult to achieve with real-world scenes.
> - **Aligned with human perception**. The ScanForgeQA dataset utilizes simplified scene scanning videos to more closely simulate how humans perceive their surroundings. In contrast, other datasets typically employ globally captured point cloud data, which does not fully reflect the selective and incremental way humans observe and understand environments.
>
> We will include this comparison in the revised manuscript and cite the relevant references to highlight the distinctions and advantages of our ScanForgeQA dataset.
>
> > **W3**: It would be better if the paper could provide some sample instances from the ScanForgeQA dataset.
>
> **A3**: Thank you for bringing this to our attention. In the supplementary materials, we included only the data construction templates and videos, but did not provide any sample instances. To address this, we have now included 20 randomly selected examples from the ScanForgeQA dataset below. Due to the rebuttal policy, we are unable to share the corresponding videos, and we apologize for any inconvenience this may cause. In the future, we will include two complete sample instances in the supplementary materials, and we will also release the constructed dataset as open source.
>
> ```
> scene_id,video_id,question_type,question,answer,options
> 39284751,39284751_1,Object Count,"How many cushions are there in the room?",3,
> 40718296,40718296_1,Room Size,"What is the size of the room in square meters?",28.5,
> 74382961,74382961_3,Relative Distance,"Which of these objects (microwave, fridge, oven) is the closest to the sink?",C,"['A. microwave', 'B. fridge', 'C. oven']"
> 29538410,29538410_3,Absolute Distance,"How far is the television from the coffee table in meters?",1.3,
> 61028437,61028437_3,Operation Feasibility,"Considering only object dimensions, is it feasible to place the printer on the bookshelf?",B,"['A. Yes', 'B. No']"
> 48971263,48971263_1,Contact Relationship,"Is there a gap between the refrigerator and the wall?",B,"['A. Yes', 'B. No']"
> 28714053,28714053_2,Object Size,"What is the length of the longest side of the table in meters?",1.8,
> 54109823,54109823_3,Room Type,"Based on object layout, what is the most likely type of this room?",B,"['A. Kitchen', 'B. Bedroom', 'C. Living Room', 'D. Bathroom']"
> 52813406,52813406_3,Object Size,"What is the size of the sofa in square meters?",2.4,
> 94278315,94278315_2,Room Type,"Is this space a living room, a kitchen, or something else?",A,"['A. Living Room', 'B. Kitchen', 'C. Bathroom', 'D. Office']"
> 10837429,10837429_1,Relative Distance,"Among the listed objects (lamp, bookshelf, plant), which one is closest to the window?",B,"['A. lamp', 'B. bookshelf', 'C. plant']"
> 87419205,87419205_1,Relative Direction,"From the viewpoint at the door facing the desk, which side is the chair on?",B,"['A. left', 'B. right', 'C. front', 'D. back']"
> 17823964,17823964_2,Operation Feasibility,"Considering only object sizes, is there enough space to put a microwave on the countertop?",A,"['A. Yes', 'B. No']"
> 82357164,82357164_1,Object Size,"What is the length of the shortest side of the bed in meters?",1.2,
> 28714053,28714053_2,Object Size,"What is the length of the longest side of the table in meters?",1.8,
> 43582170,43582170_2,Relative Direction,"If I am standing by the wardrobe and facing the mirror, which side is the lamp on?",A,"['A. left', 'B. right', 'C. front', 'D. back']"
> 57418230,57418230_2,Object Size,"How tall is the bookshelf in meters?",1.9,
> 37651942,37651942_2,Object Count,"What is the total number of towels in the room?",7,
> 92107345,92107345_1,Object Count,"How many chairs are there in the room?",4,
> 61823907,61823907_3,Contact Relationship,"Are the nightstand and the bed touching each other?",A,"['A. Yes', 'B. No']"
> 76429381,76429381_2,Absolute Distance,"What is the distance between the dining table and the sofa in meters?",2.9,
> ```
>
> > **Q1**: Regarding the ScanForgeQA dataset, are there examples of questions whose answers cannot be directly inferred from metadata alone?
>
> **A4**: No, there are no such examples in the ScanForgeQA dataset. During data construction, we first collected all scene metadata and then designed various question templates based on this metadata to ensure that every defined question is supported by the corresponding information. Furthermore, the questions in our dataset focus on fundamental attributes of objects within scenes, such as dimension, quantity, and direction, and do not involve complex question types that require extensive reasoning.

---

> > ### Comment · Reviewer_YFSW · 2025-08-04
> >
> > Thank you for the responses! It has addressed most of my concerns.

---

> > > ### Author Response · Authors · 2025-08-06
> > >
> > > We sincerely appreciate the reviewer’s thoughtful and constructive feedback. We are pleased that our responses have addressed the majority of your concerns. Should any further clarification be needed during the discussion phase, we would be happy to provide it.

---

### Decision · Program_Chairs · 2025-09-17

**Decision:**

Accept (spotlight)

**Comment:**

After the review, rebuttal and discussion all reviewers recommend acceptance. The AC agrees.

The reviewers appreciated the new data pipeline and resulting dataset.